# Anti-Inflammatory, Antioxidant, and WAT/BAT-Conversion Stimulation Induced by Novel PPAR Ligands: Results from Ex Vivo and In Vitro Studies

**DOI:** 10.3390/ph16030346

**Published:** 2023-02-24

**Authors:** Lucia Recinella, Barbara De Filippis, Maria Loreta Libero, Alessandra Ammazzalorso, Annalisa Chiavaroli, Giustino Orlando, Claudio Ferrante, Letizia Giampietro, Serena Veschi, Alessandro Cama, Federica Mannino, Irene Gasparo, Alessandra Bitto, Rosa Amoroso, Luigi Brunetti, Sheila Leone

**Affiliations:** 1Department of Pharmacy, G. d’Annunzio University, 66100 Chieti, Italy; 2Department of Clinical and Experimental Medicine, University of Messina, 98122 Messina, Italy

**Keywords:** adipose tissue, UCP1, PPAR modulators, obesity

## Abstract

Activation of peroxisome proliferator-activated receptors (PPARs) not only regulates multiple metabolic pathways, but mediates various biological effects related to inflammation and oxidative stress. We investigated the effects of four new PPAR ligands containing a fibrate scaffold—the PPAR agonists (**1a** (αEC_50_ 1.0 μM) and **1b** (γEC_50_ 0.012 μM)) and antagonists (**2a** (αIC_50_ 6.5 μM) and **2b** (αIC_50_ 0.98 μM, with a weak antagonist activity on γ isoform))—on proinflammatory and oxidative stress biomarkers. The PPAR ligands **1a-b** and **2a-b** (0.1–10 μM) were tested on isolated liver specimens treated with lipopolysaccharide (LPS), and the levels of lactate dehydrogenase (LDH), prostaglandin (PG) E_2_, and 8-iso-PGF_2_α were measured. The effects of these compounds on the gene expression of the adipose tissue markers of browning, PPARα, and PPARγ, in white adipocytes, were evaluated as well. We found a significant reduction in LPS-induced LDH, PGE_2_, and 8-iso-PGF_2_α levels after **1a** treatment. On the other hand, **1b** decreased LPS-induced LDH activity. Compared to the control, **1a** stimulated uncoupling protein 1 (UCP1), PR-(PRD1-BF1-RIZ1 homologous) domain containing 16 (PRDM16), deiodinase type II (DIO2), and PPARα and PPARγ gene expression, in 3T3-L1 cells. Similarly, **1b** increased UCP1, DIO2, and PPARγ gene expression. **2a-b** caused a reduction in the gene expression of UCP1, PRDM16, and DIO2 when tested at 10 μM. In addition, **2a-b** significantly decreased PPARα gene expression. A significant reduction in PPARγ gene expression was also found after **2b** treatment. The novel PPARα agonist **1a** might be a promising lead compound and represents a valuable pharmacological tool for further assessment. The PPARγ agonist **1b** could play a minor role in the regulation of inflammatory pathways.

## 1. Introduction

Peroxisome proliferator-activated receptors (PPARs) are important targets in metabolic diseases including obesity, metabolic syndrome, diabetes, and non-alcoholic fatty liver disease (NAFLD). PPARs are transcription factors that belong to the nuclear receptor superfamily. In particular, the activation of PPARs was found to modulate various biological effects mainly related to inflammation [1,2], oxidative stress [3], and obesity [4]. In particular, PPARs are implicated in the control of inflammatory processes induced by obesity, through their modulatory effects on the expression of proinflammatory cytokines in adipose cells [5]. PPARs have also been found to modulate the acute phase response in the liver as well as the mechanisms of inflammation in the vasculature [6,7]. There are three subtypes of PPARs, designated as PPARα, γ, and β/δ, which exhibit different tissue expression profiles and modulate specific physiological functions [5]. In this regard, a wide body of evidence suggests that PPARα plays a key role in reducing inflammation. Accordingly, mice lacking PPARα showed a prolonged inflammatory response [8,9]. Moreover, elevated levels of inflammatory markers, including vascular cell adhesion molecule-1 (VCAM-1) and serum amyloid A (SAA), were found in both mouse endothelial cells (EC) and hepatocytes lacking PPARα [10,11]. PPARγ was also shown to play a key role in the control of the inflammatory response, especially in macrophages [12]. The excessive intake of macronutrients stimulates the adipose tissue to release inflammatory mediators such as tumor necrosis factor α (TNF-α) and interleukin 6 (IL-6), and reduces the production of adiponectin [13], leading to the development of a pro-inflammatory state and oxidative stress [14,15]. On the other hand, recent studies have shown that obesity increases the risk of developing various medical conditions, including type 2 diabetes, dyslipidemia, cardiovascular diseases, and NAFLD [16,17,18]. In this context, PPARα is expressed at much higher levels in brown, relative to white, fat cells and is known to be a validated marker of brown fat cells. In addition, PPARα is involved in the activation of brown fat selective genes, such as uncoupling protein-1 (UCP1) and PR-(PRD1-BF1-RIZ1 homologous) domain containing 16 (PRDM16) [19,20].

Furthermore, PPARγ has also been shown to play a key role in inducible brown fat [21], and the activation of the browning process could be a new strategy to fight obesity and metabolic syndrome-related diseases. In this regard, PPARγ agonists were found to increase the expression of thyroid hormone activating type 2 deiodinase (DIO2), another important player in brown adipose tissue-mediated adaptive thermogenesis in animals [22].

Moreover, PPARγ has been recently suggested as a putative target for epilepsy treatment [23]. PPARγ could also be involved in the modulation of the anticonvulsant effects of EP-80317, a ghrelin receptor antagonist [24]. Finally, the antiseizure effects of cannabidiol were associated with the upregulation of PPARγ in the hippocampal CA3 region [25].

However, PPARα and PPARγ agonist drugs are known to induce side effects, including edema, weight gain, cancerogenic effects, heart failure, and renal fluid retention leading to edema [26,27,28,29,30]. Moreover, thiazolidinediones were found to reduce bone formation and stimulate bone loss in both healthy and insulin-resistant subjects [31]. These side effects have strongly restricted the clinical use of these drugs, as well as having limited the development of various PPAR ligands [32].

In previous work, we studied different PPAR agonists and antagonists in an in vitro transactivation assay. Some of these compounds showed interesting activities, with EC50 or IC50 in the low micromolar range [33,34,35,36].

In particular, we investigated the effects of PPAR ligands containing a fibrate scaffold [26], the PPAR agonists **1a** (αEC50 1.0 μM) and **1b** (γEC50 0.012 μM), and PPAR antagonists **2a** (αIC50 6.5 μM) and **2b** (αIC50 0.98 μM, with a weak antagonist activity on γ isoform) (Figure 1). Compounds **1a-b** are stilbene derivatives, with stilbene bound to the 2,2-dimethylpentanoic chain, typical of gemfibrozil [33], or a tyrosine scaffold typical of the potent and selective γ agonist GW409544 [37,38]. Compounds **2a-b** are acylsulfonamide derivatives containing a benzothiazole [39,40].

In the present study, we evaluated, by ex vivo and in vitro studies, the anti-inflammatory and antioxidant effects of these novel PPAR ligands and their role in directly activating the thermogenic program by differentiating WAT into BAT. Ligands **1a-b** and **2a-b** were tested on isolated liver specimens treated with lipopolysaccharide (LPS), a validated ex vivo experimental model of inflammation [41,42], and proinflammatory and oxidative stress biomarkers, including prostaglandin (PG) E_2_, 8-iso-PGF_2α_, and lactate dehydrogenase (LDH), were measured. In a second step, we evaluated the effects of these compounds on the gene expression of adipose tissue markers of browning (UCP1, PRDM16, and DIO2), as well as on PPARα and PPARγ in white adipocytes.

## 2. Results and Discussion

Initially, we evaluated the potential biocompatibility of the novel investigated PPAR ligands (**1a**, **1b**, **2a,** and **2b** (1 μM)) in human fibroblast (HFF-1) cells. The in vitro evaluations were conducted in both basal and LPS-induced inflammatory conditions (Figure 2 and Figure 3). The effects induced by the novel PPAR ligands were compared to those of WY-14643 (PPARα agonist; 1 μM) and pioglitazone (PPARγ agonist; 1 μM), two well-known reference compounds. WY-14643 (1 μM), pioglitazone (1 μM), **1a** (1 μM), **1b** (1 μM), **2a** (1 μM), and **2b** (1 μM), were well tolerated in human fibroblast (HFF-1) cells, in both basal and LPS-induced inflammatory conditions (Figure 2 and Figure 3).

On the basis of these results, the ex vivo experiments were carried out to evaluate the effects of compounds **1a-b** and **2a-b** in modulating LPS-induced LDH, PGE_2_, and 8-iso-PGF_2α_ production in isolated liver samples from adult male Sprague-Dawley rats, mimicking the inflammation induced by a metabolic disease [43]. In this context, we previously reported that isolated tissues that were ex vivo treated with LPS represent a validated experimental model to determine the modulatory activities of potential new drugs on inflammation and oxidative stress [44,45].

LDH is a well-known marker of tissue damage [46]. In particular, LDH production in hepatocytes increased in acute liver failure [47]. Compared to LPS treated controls, both **1a** and **1b** decreased the LPS-induced LDH activity, showing hepatoprotective effects (Figure 4). In particular, **1a** was more effective in decreasing LDH activity than the reference compound WY-14643, at the same dose, (Figure 4).

Moreover, the antagonists **2a-b** did not modify the LPS-induced LDH activity at any of the tested doses. Our findings agree with previous studies showing that PPARα agonists decreased the LDH levels in hepatic tissue [48]. Furthermore, increased LDH levels were found in liver slices from mice with a targeted deletion of PPARγ in macrophages, compared to control animals [49]. Contextually, we also found a significant reduction in LPS-induced PGE_2_ levels after the **1a** (0.1–10 μM) treatment (Figure 5).

In agreement with the present study, PPARα activation in liver samples has been demonstrated to decrease hepatic inflammation induced by an acute exposure to cytokines and other compounds [50]. The gene expression of pro-inflammatory markers, including cyclooxygenase (COX)-2, was also suppressed by PPARα agonists, in response to cytokine activation [6]. Schaefer and collaborators (2008) also showed that WY14643 inhibited the LPS-induced production of a number of pro-inflammatory mediators, including PGE_2_, and TNF-α, further confirming the beneficial effects induced by PPARα activation on tissue damage [51].

On the other hand, **1b** does not affect LPS-induced PGE_2_ production after treatment at any of the selected concentrations (Figure 5). Our findings agree with those of Yoon and collaborators (2007) showing that 15d-PGJ2, a natural ligand of PPARγ, does not reduce COX-2 gene expression and PGE_2_ production in rabbit articular chondrocytes [52]. Similarly, **2a** and **2b** did not decrease LPS-induced PGE_2_ production in liver tissues (Figure 5).

Subsequently, we studied the potential effects of the novel PPAR ligands on oxidative stress.

Oxidative stress describes the cellular damage caused by excess reactive oxygen species not adequately scavenged by antioxidants. Oxidative stress has been implicated in the development of many disorders. In this regard, the lipid peroxidation end product 8-iso-PGF_2α_ has been extensively studied as a marker of oxidative stress [53]. Therefore, following the same experimental protocol as above, we tested the effects of the new compounds on 8-iso-PGF_2α_ production. The PPARα agonist **1a** (0.1–10 μM) decreased the LPS-stimulated 8-iso-PGF_2α_ levels (Figure 6). Similarly, **1b** and **2a-b** did not exert any effect on LPS-induced 8-iso-PGF_2α_ production in liver tissues (Figure 6). Our findings showed the selective PPARα agonist **1a** as the most promising compound involved in the regulation of inflammatory and lipid peroxidation pathways.

Accordingly, PPARα has been implicated in modulating the activity of superoxide dismutase (SOD) and oxidative stress, as confirmed by the up-regulation of antioxidant enzymes such as mitochondrial SOD2 induced by PPARα activation [54]. In addition, PPARα hypothetically protects against oxidative damage in hepatocytes, developed during starvation [55], further confirming its role in modulating oxidative stress.

In addition to their role in regulating inflammatory processes, PPARs are also studied as markers of the brown adipose tissue phenotype [19]. In mammals, WAT is the largest energy reserve, while BAT has a high mitochondrial content and is known to play a key role in thermogenesis via UCP1 [13,56,57]. Variations in BAT activity could contribute to differences in energy expenditure in young and adult humans [58].

PRDM16 is a 140 kDa transcriptional co-regulator selectively expressed in brown or beige, with respect to white, adipocytes. Importantly, PRDM16 plays a critical role in modulating the differentiation-linked brown fat gene program [59,60]. Previous studies showed that the loss of PRDM16 in brown fat preadipocytes causes a loss of brown fat characteristics and induces muscle differentiation. Conversely, the ectopic expression of PRDM16 in myoblasts has stimulated brown adipogenesis [61]. The interaction with multiple DNA-binding transcriptional factors, including PPARs, is critically involved in the stimulation of brown adipogenesis induced by PRDM16 [61]. DIO2 is an enzyme playing a key role in the modulation of thyroid hormone signaling and the activation of BAT. DIO2 is also implicated as one of the major players in WAT browning [62]. Therefore, we evaluated the role of the novel PPAR ligands in the thermogenic activation of brown fat. The gene expression of adipose tissue markers of browning (UCP1, PRDM16, and DIO2), as well as PPARγ and PPARα, was evaluated after the treatment of 3T3-L1 cells, derived from 3T3 mouse cells, with testing compounds.

As shown in Figure 7, compared to control, the gene expression of UCP1 (panel A), DIO2 (panel B), and PRDM16 (panel C) were significantly enhanced, in a dose-dependent manner, when white adipocytes were incubated with **1a** at all tested doses, with the most effective dose at 10 μM. Similarly, treatment with **1b** caused a significant increase in the gene expression of UCP1 and DIO2 (Figure 7A,C) at both the 1 and 10 μM doses, with the most effective dose at 10 μM, whereas it did not change the expression of PRDM16 (Figure 7B).

Ligands **2a-b** caused a significant decrease in the gene expression of UCP1, PRDM16, and DIO2 (Figure 7A–C) when tested at 10 μM. Our present findings confirm previous studies showing that the expression of UCP1, DIO2, and PRDM16 were increased by GW7647, another PPARα agonist, in human white adipocytes [19]. Furthermore, treatment with PPARγ agonists has been shown to increase the UCP1 expression in various WAT depots, in mice [63]. UCP1 gene up-regulation is also associated with adipogenic differentiation via PPARγ or with the fatty acid oxidation required for active thermogenesis via PPARα [20]. Both PPARα and PPARγ can modulate the expression of UCP1 and both receptors are important regulators of energy balance [64]. Moreover, PRDM16 was reported to induce the thermogenic program in the subcutaneous WAT of rodents [65], and PPARα is a direct activator of PRDM16 production [19].

Furthermore, we evaluated PPAR gene expression, as reported in Figure 8.

Compared to the control, **1a** significantly increased PPARα gene expression, with a maximum effect at 10 μM (Figure 8A). **1a** also stimulated PPARγ gene expression (Figure 8B), uncorrelated with the dose range, compared to the control. These results confirm a selectivity for PPARα/γ of 1.4, as calculated for this compound and previously reported [33]. Similarly, compared to control, we found a significant increase in PPARγ gene expression (Figure 8B) after **1b** treatment in white adipocytes, at all tested doses. These results, together with the absence of PPARα gene expression, is in accordance with the previously reported data for this γ-selective agonist [37].

On the other hand, compared to the control, **2a-b** significantly decreased PPARα gene expression (Figure 8A), in the dose range 0.1–10 μM, and **2b** decreased PPARγ gene expression in a dose dependent manner (Figure 8B).

## 3. Materials and Methods

### 3.1. Chemistry

All selected compounds (**1a-b** and **2a-b**) were synthesized in the Laboratories of Medicinal Chemistry of the Department of Pharmacy, “G. d’Annunzio” University, Chieti, Italy, following procedures reported in the literature [33,34,37,40]. Melting points were determined with a Buchi Melting Point B-450 and were uncorrected. NMR spectra were recorded on a Varian Mercury 300 spectrometer with ^1^H at 300.060 MHz and ^13^C at 75.475 MHz. Proton chemical shifts were referenced to the TMS internal standard. Chemical shifts are reported in parts per million (ppm, δ units). Coupling constants are reported in units of Hertz (Hz). Splitting patterns are designed as: s, singlet; d, doublet; t, triplet; q, quartet; dd, double doublet; m, multiplet; and b, broad. Infrared spectra were recorded on a FT-IR 1600 Perkin Elmer. All commercial deuterated solvents for spectra were reagent grade and were purchased from Sigma Aldrich. The following deuterated solvents have been abbreviated: dimethyl sulfoxide (DMSO) and chloroform (CDCl_3_).

The chemical physical properties of the studied PPARs ligands are described as follows:

**1a**. 5-{4-[(*E*)-2-(4-chlorophenyl)ethenyl]phenoxy}-2,2-dimethylpentanoic acid. White solid, mp 193–194 °C. ^1^HNMR (CDCl_3_) δ 1.95 (s, 6H, CH_3_), 1.63–1.77 (m, 4H, CH_2_CH_2_), 3.94 (t, 2H, CH_2_), 6.93 (q, 2H, CH=CH), 6.84 (d, 2H, CH_Ar_), 7.27 (d, 2H, CH_Ar_), 7.36–7.41 (m, 4H, CH_Ar_); ^13^C NMR (CDCl_3_) δ 25.25 (CH_2_), 25.44 (CH_3_), 37.2 (CH_2_), 42.4 (C), 68.35 (CH_2_), 114.9 (C_Ar_), 125.4 (CH=CH), 127.6 (C_Ar_), 127.99 (CH=CH), 129.0 (C_Ar_), 129.07 (C_Ar_), 129.8 (C_Ar_), 132.8 (C_Ar_), 136.4 (C_Ar_), 159.1 (C_Ar_O), 178.1 (CO); IR (neat) 3430, 3023, 2950, 1693, 1605, 1511 cm^−1^ [33];

**1b**. 2-(((*E*)-4-Oxo-4-phenylbut-2-en-2-yl)amino)-3-(4-(3-(4-((*E*)-4-chloro styryl)phenoxy)propoxy)phenyl)propanoic acid. Amorphous yellow solid, mp 137–139 °C dec. ^1^H NMR (DMSO) δ 1.60 (s, 3H, CH_3_), 2.11 (qnt, 2H, OCH_2_CH_2_CH_2_O, J = 6.0 Hz); 2.66 (dd, 1H, PhCHH, J = 13.8 Hz, J = 8.7 Hz), 3.07 (dd, 1H, PhCHH, J = 13.8 Hz, J = 3.6 Hz), 3.83–3.90 (m, 1H, CHN), 4.05 (t, 2H, OCH_2_(CH_2_)_2_O, J = 6.0 Hz), 4.11 (t, 2H, O(CH_2_)_2_CH_2_O, J = 6.0 Hz), 5.47 (s, 1H, =CHCO), 6.79 (d, 2H, CH_Ar_, J = 9.0 Hz); 6.93 (d, 2H, CH_Ar_, J = 9.0 Hz), 7.03–7.21 (m, 4H, CH_Ar_, PhCH=CH), 7.34–7.39 (m, 5H, CH_Ar_), 7.48 (d, 2H, CH_Ar_, J = 8.7 Hz), 7.55 (d, 2H, CH_Ar_, J = 8.7 Hz), 7.73–7.77 (m, 2H, CH_Ar_), 11.35 (d, 1H, NH, J = 9.0 hz); ^13^C NMR (DMSO) δ 19.86, 27.01, 37.06, 62.03, 64.68, 64.99, 90.98, 114.73, 115.39, 127.11, 128.49, 128.61, 128.72, 129.16, 129.34, 129.61, 130.10, 131.25, 132.06, 137.07, 140.78, 141.33, 157.48, 159.05, 172.121, 184.65; IR (KBr) 3397, 2924, 2849, 1602, 1536, 1405, 1246, 831 cm^−1^ [37];

**2a**. 2-[(5-Chloro-1,3-benzothiazol-2-yl)thio]-2-phenyl-*N*-(phenylsulfonyl)acetamide. White solid, mp 140–142 °C. ^1^H NMR (CDCl_3_) δ 5.43 (s, 1H, CHS), 7.33–7.60 (m, 8H, CHAr), 7.67 (d, 1H, J = 8.4 Hz, CH_Ar_), 7.78 (d, 1H, J = 1.8 Hz, CH_Ar_), 7.91–7.97 (m, 3H, CH_Ar_), 11.05 (bs, 1H, NH); ^13^C NMR (CDCl_3_) δ 54.6 (*C*HS), 121.8, 122.1, 126.0, 126.6, 128.5, 129.0, 129.1 and 129.3 (*C*H_Ar_), 132.2, 133.0 and 133.5 (*C*_Ar_), 134.1 (*C*H_Ar_), 138.3, 152.6 and 166.7 (*C*_Ar_), 167.9 (C=O). IR (KBr) 3256, 1709, 1449, 1422, 1363, 1183 cm^−1^ [34];

**2b**. 2-[(5-chloro-1,3-benzothiazol-2-yl)thio]-*N*-({4-[(phenylacetyl)amino]phenyl}sulfonyl) pentanamide. Colourless solid, m.p. 202–204 °C (dec); ^1^H NMR (DMSO) δ 0.82 (t, 3H, J = 7.2 Hz, CH_3_CH_2_), 1.19–1.35 (m, 2H, CH_2_CH_3_), 1.72–1.85 (m, 2H, CH_2_CH), 3.65 (s, 2H, CH_2_), 4.49 (t, 1H, J = 7.2 Hz, CHS), 7.22–7.36 (m, 6H, CH_Ar_), 7.65 (d, 1H, J = 2.1 Hz, CH_Ar_), 7.71 (d, 2H, J = 9.0 Hz, CH_Ar_), 7.82 (d, 2H, J = 9.0 Hz, CH_Ar_), 7.96 (d, 1H, J = 8.4 Hz, CH_Ar_), 10.55 (bs, 1H, NH), 12.56 (bs, 1H, NH_Ar_); ^13^C NMR (DMSO) δ 14.0, 20.2, 33.9, 43.9, 51.6, 119.0, 121.2, 123.9, 125.4, 127.3, 129.0, 129.6, 129.8, 131.8, 133.0, 134.2, 136.1, 144.5, 153.7, 166.7, 169.4, 170.5. IR (KBr) 3310, 3267, 1706, 1671, 1540, 1403, 1363, 1173 cm^−1^ [40].

### 3.2. Cell Viability Assay

The cell viability was evaluated by MTT assay [3-(4,5-Dimethyl-2-thiazolyl)-2,5-diphenyl-2H-tetrazolium bromide] (Sigma, St. Louis, MO, USA) as previously described [66]. Briefly, HFF-1 cell lines were seeded in 96-well plates (5 × 10^3^ cells/well) and pretreated with 10 μg/mL lipopolysaccharide (LPS) for 24 h. Subsequently, both LPS-pretreated and non-LPS-pretreated HFF-1 cells were subjected to the PPARα agonist WY-14643 (1 μM), PPARγ agonist pioglitazone (1 μM), **1a** (1 μM), **1b** (1 μM), **2a** (1 μM), and **2b** (1 μM), or treated with a vehicle (control) for a further 24 h. After treatment, the MTT solution was added to each well and incubated at 37 °C for at least 3 h, until purple formazan crystals were formed. In order to dissolve the precipitate, the culture medium was replaced with dimethyl sulfoxide (DMSO, Euroclone). Absorbance of each well was quantified at 540 and 690 nm, using a Synergy H1 microplate reader (BioTek Instruments Inc., Winooski, VT, USA).

### 3.3. Ex Vivo Studies

Male adult Sprague–Dawley rats (200–250 g) were housed in Plexiglass cages (40 cm × 25 cm × 15 cm), with two rats per cage, in climatized colony rooms (22 ± 1 °C; 60% humidity), on a 12 h/12 h light/dark cycle (light phase: 07:00–19:00 h), with free access to tap water and food for 24 h/day throughout the study and no fasting periods. Rats were fed a standard laboratory diet (3.5% fat, 63% carbohydrate, 14% protein, and 19.5% other components without caloric value; 3.20 kcal/g). The housing conditions and experimentation procedures were strictly in agreement with the European Community ethical regulations (EU Directive no. 26/2014) on the care of animals for scientific research. In agreement with the recognized principles of “Replacement, Refinement and Reduction of Animals in Research”, liver specimens (*n* = 5 for each treatment group) were obtained as residual material from vehicle-treated rats randomized in our previous experiments approved by the local ethical committee (‘G. d’Annunzio’ University, Chieti-Pescara) and Italian Health Ministry (Project no. 885/2018-PR).

Rats were sacrificed by CO_2_ inhalation (100% CO_2_ at a flow rate of 20% of the chamber volume per min) and liver specimens were immediately collected and maintained in a humidified incubator with 5% CO_2_ at 37 °C for 4 h, in a RPMI buffer with added bacterial LPS (10 μg/mL) (incubation period), as previously reported [41].

During the incubation period, tissues were treated with the PPARα agonist WY-14643 (1 μM), and the PPARγ agonist pioglitazone (1 μM), and scalar concentrations of **1a-b** and **2a-b** (0.1–10 μM). Tissue supernatants were collected, and PGE_2_ and 8-iso-PGF_2α_ levels (ng/mg wet tissue) were measured by radioimmunoassay (RIA), as previously reported [67,68]. Briefly, specific anti-8-iso-PGF_2α_ and anti-PGE_2_ were developed in the rabbit; the cross-reactivity against other prostanoids is <0.3%. One hundred microliters of prostaglandin standard or sample were incubated overnight at 4 °C with the ^3^H-prostaglandin (3000 cpm/tube; NEN) and antibody (final dilution: 1:120,000; kindly provided by Prof. G. Ciabattoni), in a volume of 1.5 mL of 0.025 M phosphate buffer. Free and antibody-bound prostaglandins were separated by the addition of 100 μL 5% bovine serum albumin and 100 μL 3% charcoal suspension, followed by centrifuging for 10 min at 4000× *g* at 5 °C and decanting the supernatants into scintillation fluid (Ultima Gold™, Perkin Elmer, Waltham, MA, USA) for β emission counting. The detection limit of the assay method is 0.6 pg/mL. Additionally, tissue supernatants were assayed for LDH activity [44]. LDH activity was measured by evaluating the consumption of NADH in 20 mM HEPES-K+ (pH 7.2), 0.05% bovine serum albumin, 20 μM NADH, and 2 mM pyruvate using a microplate reader (excitation 340 nm, emission 460 nm) according to the manufacturer’s protocol (Sigma-Aldrich, St. Louis, MO, USA). The LDH activity was measured by evaluating the consumption of NADH in 20 mM HEPES-K+ (pH 7.2), 0.05% bovine serum albumin, 20 μM NADH, and 2 mM pyruvate using a microplate reader (excitation 340 nm, emission 460 nm) according to manufacturer’s protocol.

### 3.4. Adipocyte Culture

The 3T3-L1 cells, derived from 3T3 mouse cells, were used for the in vitro study. 3T3-L1 cells have a fibroblast-like morphology, but, under appropriate conditions, the cells differentiate into an adipocyte-like phenotype. Mouse 3T3-L1 preadipocytes were cultured in Dulbecco’s modified Eagle medium (DMEM) supplemented with 10% fetal bovine serum (FBS), 100 U/mL penicillin-streptomycin (Sigma–Aldrich, Milan, Italy) at 37 °C in a humidified atmosphere containing 5% CO_2_. Upon reaching confluence, the cells were maintained in M-1 containing glutamine (4.0 mM), sodium pyruvate (1 mM), and 10% of FBS for 1 to 2 additional days. This medium was replaced by M-2 containing M-1, insulin (1.5 µg/mL), 3-isobutyl-1-methylxanthine (IBMX) (0.5 mM), and dexamethasone (1.0 µM), for inducing adipocyte differentiation. The cells were cultured for 2 days to achieve an adipose-like phenotype. After 2 days, M-2 was replaced by M-3 containing only insulin (1.5 µg/mL), necessary for the maintenance of an adipocyte phenotype. M-3 was replaced every 2 days for 8 days; at day 8 adipocytes were treated with scalar concentrations of **1a-b** and **2a-b** (0.1–10 µM) for 24 h.

### 3.5. PCR Assay

Total mRNA was extracted from adipocytes using the Trizol LS reagent (Invitrogen, Carlsbad, CA, United States), according to the manufacturer’s protocol. Total RNA was quantified with a spectrophotometer (NanoDrop Lite, Thermo Fisher, Waltham, MA, USA) and 1 μg was reverse transcribed using the SuperScript™ IV Reverse Transcriptase (Invitrogen, Carlsbad, CA, USA) and random primers, following the manufacturer’s protocol in a volume of 20 µL. First-strand DNA (1 μL) was added to the BrightGreen qPCR Master Mix (Applied Biological Materials Inc, Richmond, BC, Canada) in a total volume of 20 µL per well to evaluate the gene expression of the target genes: UCP1, PRDM16, DIO2, PPARα, and PPARγ using specific mouse primers. For UCP1 the following primers were used: forward (ATGGTGAACCCGACAACTTC) and reverse (CAGCGGGAAGGTGATGATA). For PRDM16 the following primers were used: forward (CGAGGAGGAGACCGAAGAC) and reverse (GAAGTCTGGTGGGATTGGAA). For DIO2 the following primers were used: forward (ATGGGACTCCTCAGCGTAGA) and reverse (GGAGGAAGCTGTTCCAGACA). For DIO2 the following primers were used: forward (ATGGGACTCCTCAGCGTAGA) and reverse (GGAGGAAGCTGTTCCAGACA). For PPARα the following primers were used: forward (TCTGTCCTCTCTCCCCACTG) and reverse (CCCGGACAGCTTCCTAAGTA). For PPARγ the following primers were used: forward (CCAACTTCGGAATCAGCTCT) and reverse (CAACCATTGGGTCAGCTCTT). The samples were loaded in duplicate and GADPH was used as housekeeping gene [for GAPDH the following primers were used: forward (GTCAAGGCTGAGAATGGGAA) and reverse (ATACTCAGCACCAGCATCAC); the reaction was performed using the 2-step thermal protocol suggested by the manufacturer (Applied Biosystems, Foster City, CA, USA). The results were quantified using the 2^−ΔΔCt^ method and expressed as an n-fold increase in gene expression using untreated white adipocytes as the calibrator.

### 3.6. Statistical Analysis

The statistical analysis was performed using GraphPad Prism version 5.01 for Windows (GraphPad Software, San Diego, CA, USA). Means ± S.E.M. were determined for each experimental group and analyzed by a one-way analysis of variance (ANOVA), followed by the Newman–Keuls comparison multiple test. Statistical significance was set at *p* < 0.05.

## 4. Conclusions

In conclusion, the novel PPARα agonist **1a** might be a promising lead compound and represents a valuable pharmacological tool for further assessment, opening new perspectives on PPARα as a molecular target to afford anti-inflammatory, antioxidant, and thermogenic effects. On the other hand, the PPARγ agonist **1b** could play a minor role in the regulation of inflammatory pathways. A main limitation of our study is that we have not evaluated PPAR antagonists as reference compounds. Further studies are needed to accurately evaluate the in vivo activities of these compounds.

## Figures and Tables

**Figure 1 pharmaceuticals-16-00346-f001:**
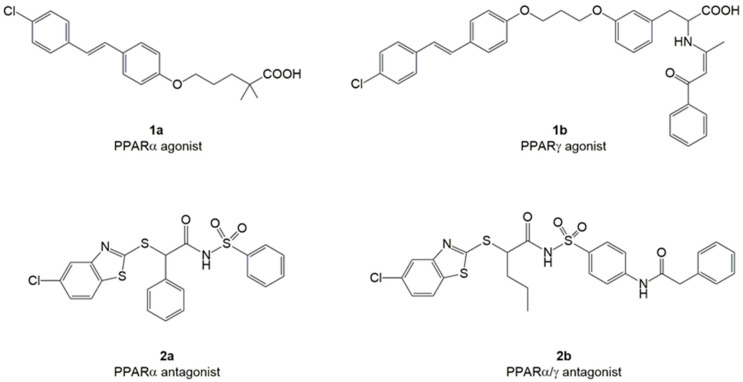
The chemical structures of PPAR agonists (**1a-b**) and antagonists (**2a-b**).

**Figure 2 pharmaceuticals-16-00346-f002:**
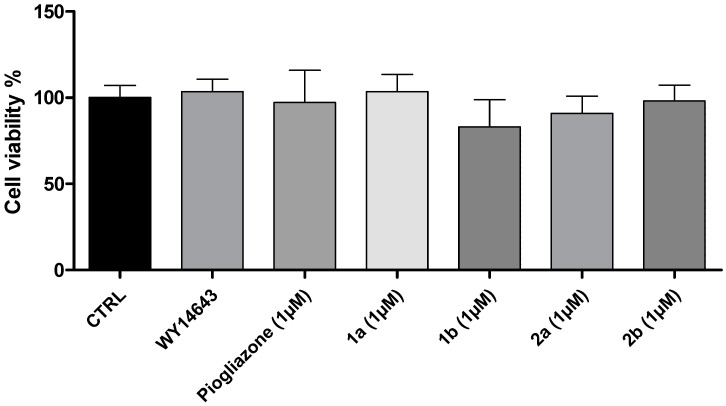
An MTT assay of human fibroblast (HFF-1) cells treated with WY-14643 (1 μM), pioglitazone (1 μM), **1a** (1 μM), **1b** (1 μM), **2a** (1 μM), and **2b** (1 μM), for 24 h, in basal conditions. Data were reported as means ± SEM.

**Figure 3 pharmaceuticals-16-00346-f003:**
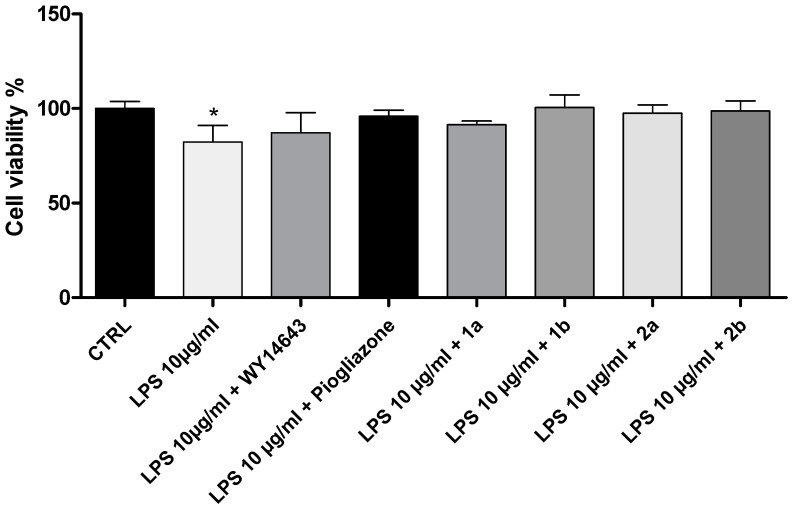
An MTT assay of LPS-pretreated human fibroblast (HFF-1) cells, exposed to WY-14643 (1 μM), pioglitazone (1 μM), **1a** (1 μM), **1b** (1 μM), **2a** (1 μM), and **2b** (1 μM), for 24 h. Data were reported as means ± SEM. The results were analyzed by analysis of variance (ANOVA) followed by the Newman–Keuls post-hoc test. ANOVA, *p* < 0.01; * *p* < 0.05 vs. CTRL.

**Figure 4 pharmaceuticals-16-00346-f004:**
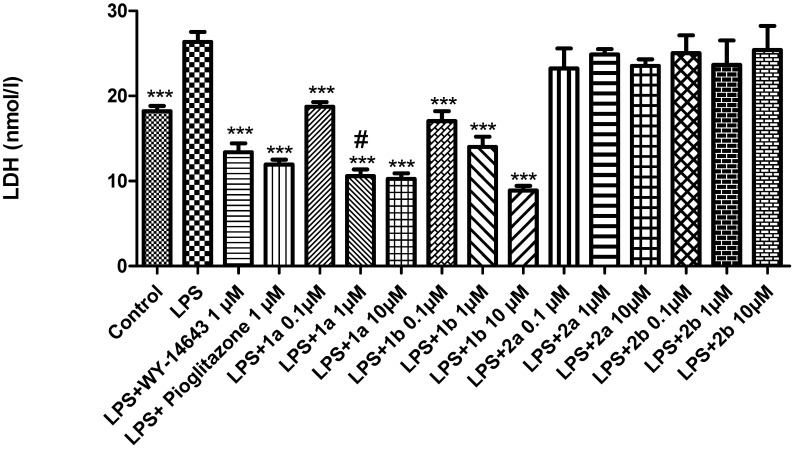
The effects of WY-14643 (1 μM), pioglitazone (1 μM), and **1a-b** and **2a-b** (0.1–10 μM), on the LPS-induced LDH activity (nmol/l) in rat liver, *ex vivo*. Data were reported as means ± SEM. The results were analyzed by analysis of variance (ANOVA) followed by the Newman–Keuls post-hoc test. ANOVA, *p* < 0.0001; *post-hoc* *** *p* < 0.001 vs. LPS-treated group; # *p* < 0.05 vs. LPS + WY-14643.

**Figure 5 pharmaceuticals-16-00346-f005:**
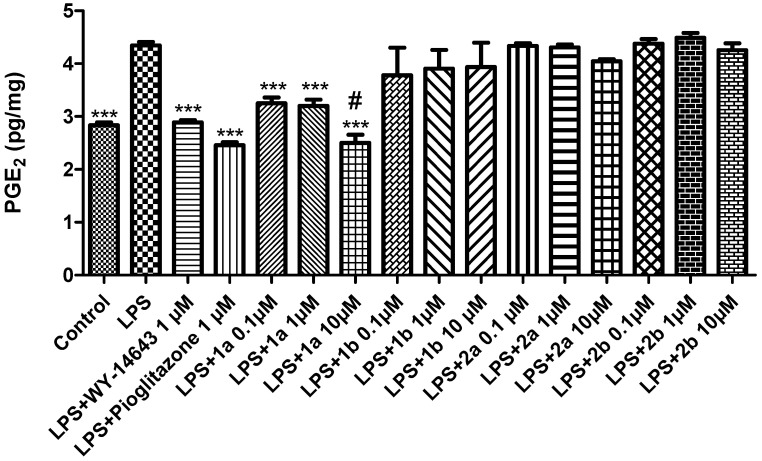
The effects of WY-14643 (1 μM), pioglitazone (1 μM), and **1a-b** and **2a-b** (0.1–10 μM), on the LPS-induced production of PGE_2_ (pg/mg) in rat liver, ex vivo. Data were reported as means ± SEM. The results were analyzed by analysis of variance (ANOVA) followed by the Newman–Keuls post-hoc test. ANOVA, *p* < 0.0001; post-hoc *** *p* < 0.001 vs. LPS-treated group; # *p* < 0.05 vs. LPS + WY-14643.

**Figure 6 pharmaceuticals-16-00346-f006:**
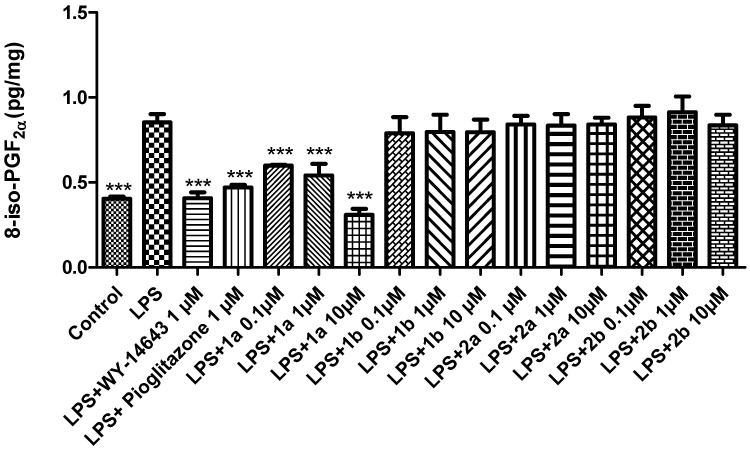
The effects of WY-14643 (1 μM), pioglitazone (1 μM), and **1a-b** and **2a-b** (0.1–10 μM), on the LPS-induced production of 8-iso-PGF_2α_ (pg/mg) in rat liver, ex vivo. Data were reported as means ± SEM. The results were analyzed by analysis of variance (ANOVA) followed by the Newman–Keuls post-hoc test. ANOVA, *p* < 0.0001; post-hoc *** *p* < 0.001 vs. LPS-treated group.

**Figure 7 pharmaceuticals-16-00346-f007:**
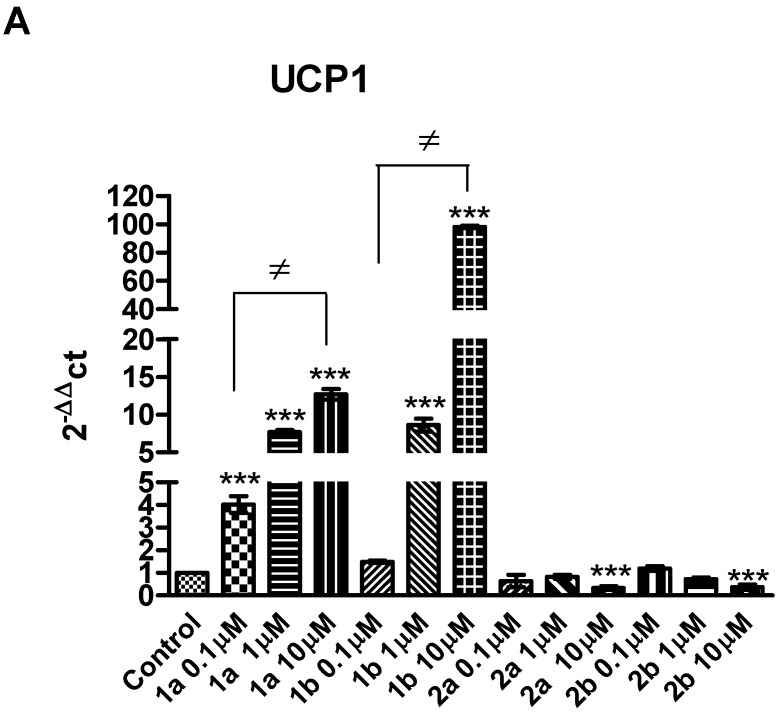
The effects of **1a-b** and **2a-b** (0.1–10 μM), on UCP1 (**A**), PRDM16 (**B**), and DIO2 (**C**), gene expression in white adipocytes. Data were reported as means ± SEM. The results were analyzed by analysis of variance (ANOVA) followed by the Newman–Keuls post-hoc test. ANOVA, *p* < 0.0001; post-hoc *** *p* < 0.001 vs. control group; ≠ *p* < 0.05 vs. **1a** (0.1,1 μM); ≠ *p* < 0.05 vs. **1b** (0.1,1 μM).

**Figure 8 pharmaceuticals-16-00346-f008:**
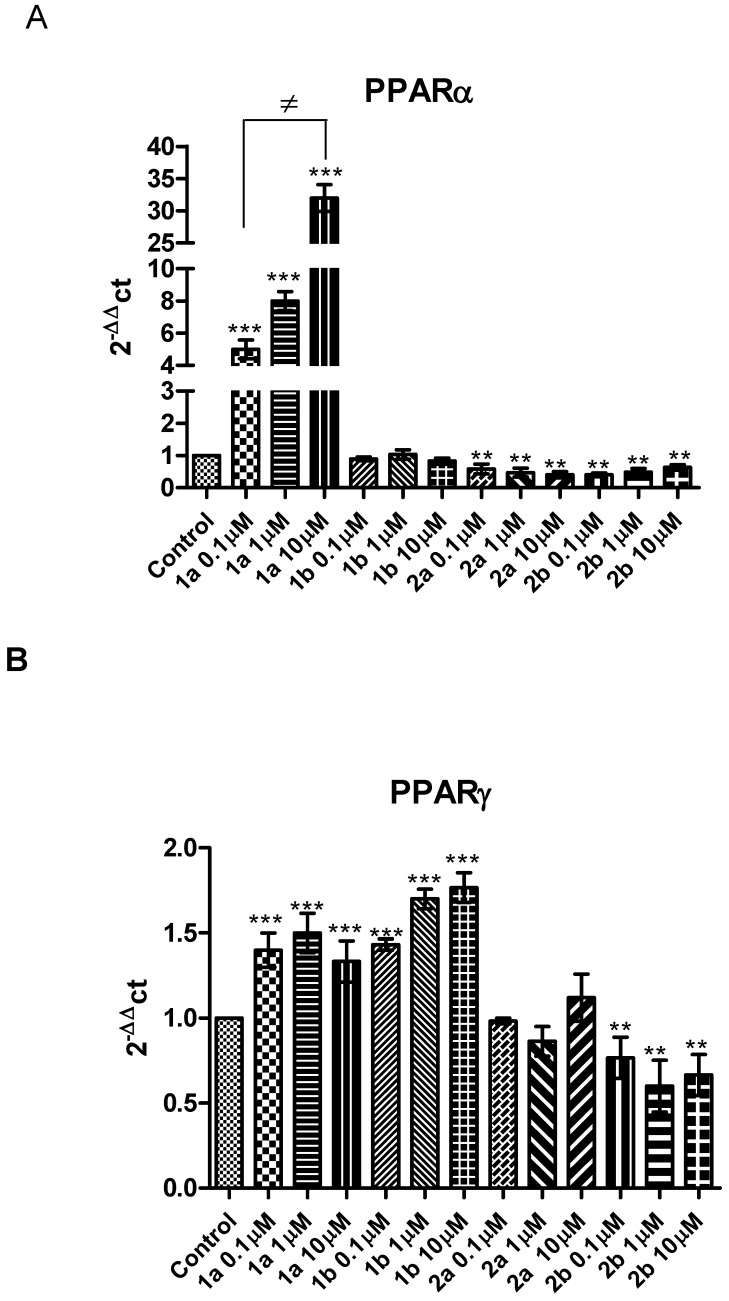
The effects of **1a-b** and **2a-b** (0.1–10 μM), on PPARγ (**A**) and PPARα (**B**) gene expression in white adipocytes. Data are reported as means ± SEM. The results were analyzed by analysis of variance (ANOVA) followed by the Newman–Keuls post-hoc test. ANOVA, *p* < 0.0001; post-hoc ** *p* < 0.01, *** *p* < 0.001 vs. control group; ≠ *p* < 0.05 vs. **1a** (0.1–1 μM).

## Data Availability

Data is contained within the article.

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
