# Peer review of "Anti-Inflammatory, Antioxidant, and WAT/BAT-Conversion Stimulation Induced by Novel PPAR Ligands: Results from Ex Vivo and In Vitro Studies"

_pharmaceuticals, 2023, doi:10.3390/ph16030346_

Round 1

Reviewer 1 Report

The manuscript by Recinella et al. about the beneficial effects induced by novel PPARs ligands is of interest. However, the following points need to be deeply improved:

Abstract

1)      Lines 17-20: The sentence is not clear. Please rephrase.

Introduction

2)      The characteristics and role of each subtype of PPARs are quite well summarized. However, as the manuscript is focused on PPARs agonists and antagonists, I would have expected an introduction possibly summarizing the pros and cons of the compounds already available and the current gap that pushed you to research new PPARs agonists and antagonists. Thus, the introduction should be modified.

3)      It could be highlighted that PPAR agonists could have a possible application in the treatment of much many diseases of those already well-known. For instance, it was recently published that the PPARγ could be a putative target for epilepsy treatment (Senn et al., 2023 doi.org/10.1016/j.pharmthera.2022.108316). Moreover, it was found that PPARγ might be a useful marker of response to the anticonvulsant EP-80317. Finally, the antiseizure effects of cannabidiol were associated with upregulation of PPARγ in the hippocampal CA3 region (Costa et al., 2022 doi.org/10.3390/ph15050495).

Results and Discussion

4)      Why did you use any PPARs antagonists as reference compounds?

Materials and Methods

5)      Line 368: Please clarify the meaning of the abbreviations in the formula.

Conclusions

6)      The limitations of the study should be reported.

Author Response

Reviewer 1:

The manuscript by Recinella et al. about the beneficial effects induced by novel PPARs ligands is of interest. However, the following points need to be deeply improved:

Abstract

  • Lines 17-20: The sentence is not clear. Please rephrase.

Response to Reviewer: We thank the Reviewer for the comment. In the Revised Manuscript, we performed the kindly suggested correction (Lines 18-21 of the Revised Manuscript).

Introduction

  • The characteristics and role of each subtype of PPARs are quite well summarized. However, as the manuscript is focused on PPARs agonists and antagonists, I would have expected an introduction possibly summarizing the pros and cons of the compounds already available and the current gap that pushed you to research new PPARs agonists and antagonists. Thus, the introduction should be modified.

Response to Reviewer: Thank you for your suggestion. In the Revised Manuscript, we performed the kindly suggested correction (Lines 74-79 of the Revised Manuscript).

  • It could be highlighted that PPAR agonists could have a possible application in the treatment of much many diseases of those already well-known. For instance, it was recently published that the PPARγ could be a putative target for epilepsy treatment (Senn et al., 2023 doi.org/10.1016/j.pharmthera.2022.108316). Moreover, it was found that PPARγ might be a useful marker of response to the anticonvulsant EP-80317. Finally, the antiseizure effects of cannabidiol were associated with upregulation of PPARγ in the hippocampal CA3 region (Costa et al., 2022 doi.org/10.3390/ph15050495).

Response to Reviewer: We thank the Reviewer for the insightful suggestion. In the Revised Manuscript, we have inserted some informations about the possible application in the treatment of epilepsy (Lines 70-73 of the Revised Manuscript).

Results and Discussion

  • Why did you use any PPARs antagonists as reference compounds?

Response to Reviewer: We thank the Reviewer for the comment. We apologize for the inconvenience. In our study we have not included any PPAR antagonists as reference compounds and this could be the main limitation of our study. At this regard, in the Conclusions we added the following sentence: “A main limitation of our study is that we have not evaluated used any PPAR antagonist as reference compound” (Lines 414-415 of the Revised Manuscript).

Materials and Methods

  • Line 368: Please clarify the meaning of the abbreviations in the formula.

Response to Reviewer: Sorry but we did not find any abbreviation in the formula, at line 368. Could you specify what is the abbreviation to clarify?

Conclusions

  • The limitations of the study should be reported.

Response to Reviewer: Thank you for your comment. In the Conclusions we added the following sentence: “A main limitation of our study is that we have not evaluated used any PPAR antagonist as reference compound” (Lines 414-415 of the Revised Manuscript).

Reviewer 2 Report

In the manuscript entitled “Beneficial effects induced by novel PPARs ligands: results from ex vivo and in vitro studies”, the authors investigated the anti-inflammatory, antioxidant and WAT/BAT conversion stimulation of four PPAR modulators [1 PPARα and 1 PPARγ agonists; and 2 PPARα antagonists (one of them has a weak inhibitory effect on PPARγ)]. The manuscript is clear, but it needs some careful language revision/editing. Below are some comments that may hopefully help improve the manuscript.

1.      The title is too vague, it does not say what kind of beneficial effects these ligands have. Please modify and make it clearer.

2.      The  authors studied how their ligand affect WAT to BAT differentiation in control adipocytes that are not under inflammatory influence. How does this compare to the in vivo case of obese patients where adipose tissue is expected to release inflammatory mediators? Do you expect stimulation of browning of adipose tissue with these ligands? Please discuss showing how this process (WAT/BAT conversion) is usually affected by obesity and how your ligands might affect it.

3.      P.4 line, 116: “Compared to control, both 1a and 1b decreased LPS-induced LDH activity, showing hepatoprotective effects (Figure 4).” Should this be compared to LPS? Please revise.

4.      How many livers were used in the ex vivo study? And please describe how they were divided between the treatments.

5.      P.9 line 204: “…. PRDM16 (Figure 7, panel B).” should panel B be panel C? please revise.

6.      P. 12 line 311: “….. (10 μg/mL) (incubation period), as previously reported [31].” Please state what was the incubation period?

7.      P. 13 line 357 “………using specific mouse primers.” Please provide primer sequence for all genes.

8.      P. 13 line 367 “The number of animals randomized for each experimental group was calculated on the basis of the “Resource Equation” N = (E + T)/T (10 ≤ E ≤ 20) [59], according to the guidelines suggested by the National Centre for the Replacement, Refinement and Reduction of Animals in Research (NC3RS) and reported on the following web site:https://www.nc3rs.org.uk/experimental-designstatistics.” There is no animal study, please remove this whole sentence.

Author Response

Dear Editor,

thanks for the questions posed by the Reviewers.

We have revised our manuscript (Manuscript ID: pharmaceuticals-2221573, Title: Beneficial effects induced by novel PPARs ligands: results from ex vivo and in vitro studies”) and the questions posed by the Reviewers have been responded point by point. The changes have been highlighted in yellow in the revised version of the manuscript.

Reviewer 2:

Comments and Suggestions for Authors

In the manuscript entitled “Beneficial effects induced by novel PPARs ligands: results from ex vivo and in vitro studies”, the authors investigated the anti-inflammatory, antioxidant and WAT/BAT conversion stimulation of four PPAR modulators [1 PPARα and 1 PPARγ agonists; and 2 PPARα antagonists (one of them has a weak inhibitory effect on PPARγ)]. The manuscript is clear, but it needs some careful language revision/editing. Below are some comments that may hopefully help improve the manuscript.

  1. The title is too vague, it does not say what kind of beneficial effects these ligands have. Please modify and make it clearer.

Response to Reviewer: As kindly suggested by the Reviewer, the Revised Manuscript was entitled “Anti-inflammatory, antioxidant and WAT/BAT conversion stimulation induced by novel PPARs ligands: results from ex vivo and in vitro studies”.

  1. The authors studied how their ligand affect WAT to BAT differentiation in control adipocytes that are not under inflammatory influence. How does this compare to the in vivo case of obese patients where adipose tissue is expected to release inflammatory mediators? Do you expect stimulation of browning of adipose tissue with these ligands? Please discuss showing how this process (WAT/BAT conversion) is usually affected by obesity and how your ligands might affect it.

Response to Reviewer: PPARs have been suggested to act as critical transcription factors in regulation of adipose tissue development and function, as well as in the conversion of white into brown-like adipocytes.  Recently, Kroon and collaborators (2020) showed that selective PPARγ activation induced a moderate  browning of WAT, while dual PPARα/γ activation was found exert a marked browning, in obese mice (Kroon T, Harms M, Maurer S, Bonnet L, Alexandersson I, Lindblom A, Ahnmark A, Nilsson D, Gennemark P, O'Mahony G, Osinski V, McNamara C, Boucher J. PPARγ and PPARα synergize to induce robust browning of white fat in vivo. Mol Metab. 2020 Jun;36:100964). In addition, stimulation of WAT browning has been proposed as potential therapeutic strategy in treatment of obesity and type 2 diabetes (Peng XR, Gennemark P, O'Mahony G, Bartesaghi S. Unlock the Thermogenic Potential of Adipose Tissue: Pharmacological Modulation and Implications for Treatment of Diabetes and Obesity. Front Endocrinol (Lausanne). 2015 Nov 19;6:174.). On the basis of our present findings, we hypothesized that our ligands 1a and 1b could induce browning of white fat, in obesity.  

  1. P.4 line, 116: “Compared to control, both 1a and 1b decreased LPS-induced LDH activity, showing hepatoprotective effects (Figure 4).” Should this be compared to LPS? Please revise.

Response to Reviewer: We thank the Reviewer for the comment. In the Revised Manuscript, we performed the suggested correction (Line 132 of the Revised Manuscript).  

  1. How many livers were used in the ex vivo study? And please describe how they were divided between the treatments.

Response to Reviewer: We thank the Reviewer for the comment. In the Revised Manuscript, we performed the suggested correction (Line 332 of the Revised Manuscript). 

  1. P.9 line 204: “…. PRDM16 (Figure 7, panel B).” should panel B be panel C? please revise.

Response to Reviewer: We thank the Reviewer for the comment. In the Revised Manuscript, we performed the suggested correction (Lines 221-222 of the Revised Manuscript). 

  1. P. 12 line 311: “….. (10 μg/mL) (incubation period), as previously reported [31].” Please state what was the incubation period?

Response to Reviewer: We thank the Reviewer for the comment. In the Revised Manuscript, we specified that liver specimens were immediately collected and maintained in humidified incubator with 5% CO2 at 37 °C for 4 h, in RPMI buffer with added bacterial LPS (10 μg/mL) (incubation period) (Line 338 of the Revised Manuscript).   

  1. P. 13 line 357 “………using specific mouse primers.” Please provide primer sequence for all genes.

Response to Reviewer: We thank the Reviewer for the comment. In the Revised Manuscript, we provided primer sequence for all genes (Lines 386-398 of the Revised Manuscript).

  1. P. 13 line 367 “The number of animals randomized for each experimental group was calculated on the basis of the “Resource Equation” N = (E + T)/T (10 ≤ E ≤ 20) [59], according to the guidelines suggested by the National Centre for the Replacement, Refinement and Reduction of Animals in Research (NC3RS) and reported on the following web site:https://www.nc3rs.org.uk/experimental-designstatistics.” There is no animal study, please remove this whole sentence.

Response to Reviewer: We thank the Reviewer for the comment. In the Revised Manuscript, we deleted the whole sentence.

Round 2

Reviewer 1 Report

Materials and Methods

  • Line 368: Please clarify the meaning of the abbreviations in the formula.

Response to Reviewer: Sorry but we did not find any abbreviation in the formula, at line 368. Could you specify what is the abbreviation to clarify?

Response to authors: the cited formula is: N = (E + T)/T (10 ≤ E ≤ 20). However, these sentences were deleted in the new version "The number of animals randomized for each experimental group was calculated on the basis of the “Resource Equation” N = (E + T)/T (10 ≤ E ≤ 20) [59], according to the guidelines suggested by the National Centre for the Replacement, Refinement and Reduction of Animals in Research (NC3RS) and reported on the following web site: https://www.nc3rs.org.uk/experimental-designstatistics.

Author Response

Comments and Suggestions for Authors

Materials and Methods

  • Line 368: Please clarify the meaning of the abbreviations in the formula.

Response to Reviewer: Sorry but we did not find any abbreviation in the formula, at line 368. Could you specify what is the abbreviation to clarify?

Response to authors: the cited formula is: N = (E + T)/T (10 ≤ E ≤ 20). However, these sentences were deleted in the new version "The number of animals randomized for each experimental group was calculated on the basis of the “Resource Equation” N = (E + T)/T (10 ≤ E ≤ 20) [59], according to the guidelines suggested by the National Centre for the Replacement, Refinement and Reduction of Animals in Research (NC3RS) and reported on the following web site: https://www.nc3rs.org.uk/experimental-designstatistics.

Response to Reviewer: We thank the Reviewer for the elucidation. As required by the Reviewer 2, in the Revised Manuscript, we have deleted these sentences.